# Cooperative Task Assignment of a Heterogeneous Multi-UAV System Using an Adaptive Genetic Algorithm

**Fang Ye, Jie Chen** [ID] **, Yuan Tian** *[ID] **and Tao Jiang**

College of Information and Communication Engineering, Harbin Engineering University, Harbin 150001, China; yefang0923@126.com (F.Y.); sandra@hrbeu.edu.cn (J.C.); jiangtao_heu@126.com (T.J.)
* Correspondence: tianyuan347@126.com

**Abstract:** The cooperative multiple task assignment problem (CMTAP) is an NP-hard combinatorial optimization problem. In this paper, CMTAP is to allocate multiple heterogeneous fixed-wing UAVs to perform a suppression of enemy air defense (SEAD) mission on multiple stationary ground targets. To solve this problem, we study the adaptive genetic algorithm (AGA) under the assumptions of the heterogeneity of UAVs and task coupling constraints. Firstly, the multi-type gene chromosome encoding scheme is designed to generate feasible chromosomes that satisfy the heterogeneity of UAVs and task coupling constraints. Then, AGA introduces the Dubins car model to simulate the UAV path formation and derives the fitness value of each chromosome. In order to comply with the chromosome coding strategy of multi-type genes, we designed the corresponding crossover and mutation operators to generate feasible offspring populations. Especially, the proposed mutation operators with the state-transition scheme enhance the stochastic searching ability of the proposed algorithm. Last but not least, the proposed AGA dynamically adjusts the number of crossover and mutation populations to avoid the subjective selection of simulation parameters. The numerical simulations verify that the proposed AGA has a better optimization ability and convergence effect compared with the random search method, genetic algorithm, ant colony optimization method, and particle search optimization method. Therefore, the effectiveness of the proposed algorithm is proven.

**Keywords:** multi-UAV system; task assignment; adaptive genetic algorithm; state-transition strategy; Dubins car model

## 1. Introduction

The increasing complexity of the electromagnetic environment and the integration of weapons systems in modern battlefields have brought unprecedented challenges to the mission execution of aerial vehicles (AV) [1,2]. Due to the low cost, zero casualty, and good flexibility of unmanned aerial vehicles (UAVs), UAVs have gradually replaced the role of AV in carrying out dangerous and harsh tasks [3,4]. However, tasks in the modern battlefield become more diverse and complex. A single UAV with limited capacity cannot ensure the completion of a mission [5,6]. Thus, it is an inevitable trend to study the cooperation of multiple UAVs [7].

The cooperative multiple task assignment problem (CMTAP) is a core topic of multi-UAV cooperative operation. Effective cooperative task assignment can develop a mission plan for a multi-UAV system when satisfying the operational requirements. In this paper, the CMTAP model is to allocate multiple heterogeneous fixed-wing UAVs to perform multiple tasks (classify, attack, and verify) against multiple stationary ground targets in the suppression of enemy air defense (SEAD)

scenario. Many scholars focused on how to generate the CMTAP solution through maximizing global benefits or minimizing global costs.

Some cooperative control and decision-making methods like mixed integer linear programming (MILP) [8], the tree search method [9], the simulated annealing method (SA) [10], the taboo search method (TS) [11], the differential evolution algorithm (DE) [12], ant colony optimization (ACO) [13,14], particle search optimization (PSO) [15,16], and the genetic algorithm (GA) [17,18] have been applied to solve CMTAP. As is known to all, CMTAP is an NP-hard combinatorial optimization problem. The exhaustive enumerations of MILP and the tree search method lead to prohibitive computational complexity when handling large and complex missions [8,9]. Hence, the application of intelligent optimization methods is more conducive to solving CMTAP. The work in [10] proposed an improved K-means clustering algorithm of SA to solve the multi-UAV task allocation in the cooperative reconnaissance scenario. CMTAP with a time window was built in [11], followed by a modified TS. In [11], the attack UAV could only launch one projectile, which does not meet the actual demand. To solve CMTAP, an adaptive DE based on an adaptive factor and population catastrophe was proposed in [12]. Taking CMTAP as a vehicle routing problem, the work in [13] presented a revised ACO to solve the multi-UAV task allocation and route planning. An improved PSO was also put forward in [15] to solve CMTAP. However, swarm intelligence algorithms like ACO and PSO may produce a larger number of infeasible individuals during the population iterations [19]. The application of swarm intelligence algorithms to CMTAP requires extra computation to solve the infeasible individuals.

GA has strong universality, a simple encoding strategy, and genetic operators, and it is widely applied in machine learning, pattern recognition, etc. As a stochastic optimization searching method, GA simulates the biological evolution mechanism of nature (survival of the fittest). On the one hand, GA directly takes the objective function as the searching information, which has strong robustness [20]. On the other hand, GA balances the global search and local search with a proper encoding strategy and genetic operators [21]. Thus, GA has unique inherent parallelism and global optimization ability, which is suitable for CMTAP.

Considering the point targets, line targets, and area targets in the cooperative reconnaissance scenario, the work in [22] raised the opposition-based GA using double-chromosomes encoding. By defining the visitation angle discretization of the UAV over the target, the work in [23] presented the integrated scheme of task assignment and motion planning based on GA and graph representation. Besides, the work in [24] used binary matrices to define the chromosomes of GA, which may lead to inevitable computational complexity in large scenarios. The work in [19] put forward the modified GA based on multi-type genes and mirror representation of UAVs. However, the presented GA in [19] may generate deadlock chromosomes during the population iterations. In [25], GA was developed to find the optimal solution for the CMTAP model, where the Dubins car model was used to establish the UAVs' path formation.

The focus of this paper is to solve the CMTAP of heterogeneous fixed-wing UAVs performing an SEAD mission on multiple stationary ground targets. Both the heterogeneity of UAVs and task coupling constraints are considered. Although GA has been widely applied to solve CMTAP, these existing methods cannot be directly applied under these assumptions. Therefore, we propose an adaptive genetic algorithm (AGA). Firstly, the multi-type genes are introduced to establish a deadlock-free chromosome encoding strategy. The proposed chromosome encoding strategy ensures that the generated chromosomes satisfy the UAV heterogeneity and task coupling constraints. To calculate the mission execution time, the Dubins car model is used to simulate the UAV path formation. According to the chromosome encoding strategy, we modified the crossover and mutation operators to guarantee the feasibility of the GA population. Besides, the proposed AGA dynamically adjusts the number of crossover and mutation populations based on the iteration time.

The structure of this paper is as follows. Section 2 presents the problem statements of the CMTAP model. Section 3 elaborates on the proposed algorithm, including the chromosome encoding strategy, the calculation of the fitness, the design of crossover and mutation operators, and the adaptive setting

of the crossover and mutation populations. Simulations and analyses are expressed in Section 4. Finally, Section 5 concludes this paper.

## 2. Problem Statements

This paper concentrates on the cooperative multiple task assignment problem (CMTAP) with UAVs' heterogeneity and task coupling constraints.

### 2.1. Task Coupling Constraints

Multiple stationary ground targets are considered in the SEAD mission. The target set is defined as:

$$\mathbf{T} = \{T_1, T_2, \ldots, T_{N_T}\} \tag{1}$$

where $N_T$ is the number of targets.

In the SEAD mission, classify, attack, and verify tasks need to be performed on each target in sequence. The task set is:

$$M_T = \{C, A, V\} \tag{2}$$

where $C, A, V$ separately represent the classify, attack, and verify tasks.

The task coupling constraints reflect two aspects:

(1) The SEAD mission of a certain target is accomplished only if $C, A, V$ tasks are all performed. According to Equations (1) and (2), the number of executed tasks on each target is $|M_T| = 3$, and the total number of tasks is $N_t = N_T \cdot |M_T| = 3N_T$.

(2) The performing order of $C, A, V$ tasks follows strict task precedence constraints. Task $A$ can only be performed after the target is classified, and task $V$ can only be executed after completing task A.

### 2.2. UAVs' Heterogeneity

To achieve the SEAD mission on targets, surveillance, combat, and munition UAVs are considered in the multi-UAV system. The UAV set is denoted as:

$$\mathbf{U} = \{U_1, U_2, \ldots, U_{N_U}\} \tag{3}$$

where $N_U$ is the number of UAVs.

The heterogeneity of UAVs reflect two parts:

(1) Different UAVs have different capabilities. Surveillance can only perform the classify and verify tasks; munition UAVs can only perform the attack tasks; and combat UAVs can perform all tasks. Accordingly, we denote the UAV sets with different capabilities as:

$$\mathbf{U}_S = \{ \begin{array}{l} U_1, U_2, \ldots, U_{N_{U_S}}, U_{N_{N_S}+1}, \\ U_{N_{N_S}+2}, \ldots, U_{N_{N_S}+N_{U_C}} \end{array} \} \tag{4}$$

$$\mathbf{U}_A = \{ \begin{array}{l} U_1, U_2, \ldots, U_{N_{U_M}}, U_{N_{N_M}+1}, \\ U_{N_{N_M}+2}, \ldots, U_{N_{U_M}+N_{U_C}} \end{array} \} \tag{5}$$

where $N_{U_S}, N_{U_C}, N_{U_M}$ separately represent the number of surveillance, combat, and munition UAVs.

Apparently, $N_U = N_{U_S} + N_{U_C} + N_{U_M}$. $\mathbf{U}_S, \mathbf{U}_A$ separately define the UAVs that are capable of performing the surveillance $(C, V)$ and attack $(A)$ tasks.

(2) Different UAVs have different kinematic parameters, including different cruise speeds $[v_1, v_2, \ldots, v_{N_U}]$ and turning radii $[r_1, r_2, \ldots, r_{N_U}]$. The Dubins car model [26] is adopted to describe the kinematic path of UAVs. Given two points with known orientations, the Dubins car model can generate the shortest UAV path. The possible Dubins paths are shown in Figure 1.

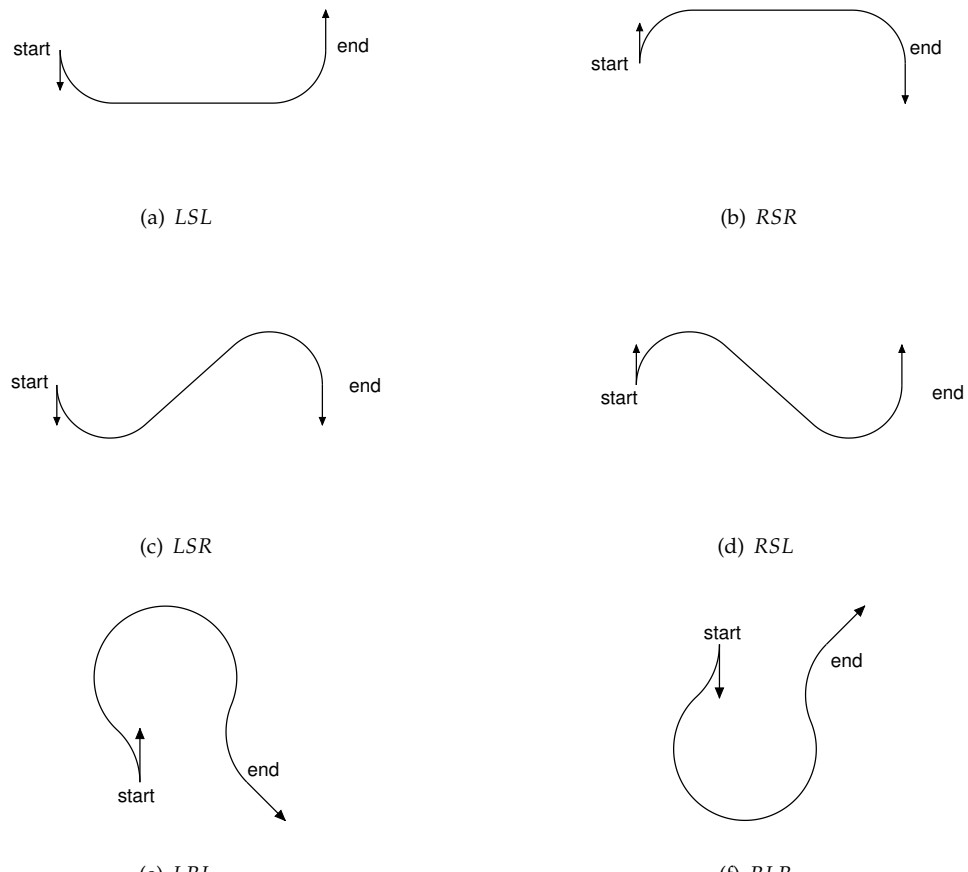

**Figure 1.** Examples of Dubins paths.

We can see from Figure 1 that the Dubins path of the UAV must be one of the following six combination of line segments and curvature arcs $[LSL, RSR, LSR, RSL, LRL, RLR]$, where $R$ is the clockwise turn, $L$ is the counterclockwise turn, and $S$ is the straight line [27]. Using the cruise speed and turning radius of the UAV, we can derive the cruise time of the UAV between two configurations based on its Dubins path.

### 2.3. CMTAP Model

CMTAP is to allocate heterogeneous UAVs to perform the SEAD mission on multiple targets under the constraints of the UAVs' heterogeneity and task coupling. The objective of the CMTAP model is to minimize the execution time that the multi-UAV system takes to realize the whole SEAD mission.

$$\min J = \max_{U_k \in \mathbf{U}} t_{U_k} \tag{6}$$

where $t_{U_k}$ is the execution time of UAV $U_k$ performing all its assigned tasks.

$$t_{U_k} = \sum_{i=1}^{N_V} \sum_{j=1}^{N_T} \sum_{m=1}^{3} \left\{ \frac{X_{(q_i,q_j)}^{U_k,m} d_{(q_i,q_j)}^{U_k}}{v_k} \right\} \tag{7}$$

where $X_{(q_i,q_j)}^{U_k,m}$ is the binary decision variable. $X_{(q_i,q_j)}^{U_k,m} = 1$ means that UAV $U_k$ flies from $q_i$ to $q_j$ to perform task $m$ on target $T_j$; otherwise, $X_{(q_i,q_j)}^{U_k,m} = 0$. $d_{(q_i,q_j)}^{U_k}$ is the Dubins path of UAV $U_k$ flying from $q_i$ to



$q_j$. $q_i, q_j$ separately represent the last and current configurations of UAV $U_k$. $N_V = N_U + N_T$ represents all possible configurations of UAVs during the mission, including their take-off configurations and possible configurations on targets.

The first constraint of the CMTAP model is that $X^{U_k,m}_{(q_i,q_j)}$ should satisfy the UAVs' capability defined in Section 2.2. The UAV $U_k$ should have the corresponding capability for its assigned task $m$ on target $T_j$.

Other constraints of the CMTAP model are:

$$\sum_{k=1}^{N_U} \sum_{i=1}^{N_V} X^{U_k,m}_{(q_i,q_j)} = 1, \forall m = 1, 2, 3 \tag{8}$$

$$\sum_{k=1}^{N_U} \sum_{i=1}^{N_V} \sum_{m=1}^{3} X^{U_k,m}_{(q_i,q_j)} = |M_T| \tag{9}$$

$$t^C_{T_j} < t^A_{T_j} < t^V_{T_j} \tag{10}$$

where $\forall T_j \in \mathbf{T}$.

It is easily checked that Equations (8)–(10) denote two aspects of the task coupling constraints described in Section 2.1. Firstly, Equation (8) reflects that each task should be assigned just once, and Equation (9) shows that the number of tasks performed on each target should be exactly $|M_T| = 3$. Then, Equation (10) expresses that the execution of $C, A, V$ tasks should follow the strict task precedence constraint.

## 3. Proposed Algorithm

Since the CMTAP model is an NP-hard combinatorial optimization problem, we propose an adaptive genetic algorithm (AGA) with a multi-type gene chromosome encoding strategy. As key elements of GA, the chromosome encoding, calculation of fitness, and genetic operations are modified in this paper. Then, the adaptive settings of the number of crossover and mutation populations during the GA iteration are also raised.

### 3.1. Chromosome Encoding Strategy

To describe the multiple tasks in the SEAD mission, the chromosome encoding strategy with multi-type genes is put forward.

#### 3.1.1. Chromosome with Multi-Type Genes

Suppose that multi-UAV system $\mathbf{U} = \{U_1^S, U_2^C, U_3^M\}$ from the same base needs to perform the SEAD mission on targets $\mathbf{T} = \{T_1, T_2\}$. One feasible chromosome with multi-type genes is shown in Figure 2.

In Figure 2, the chromosome contains $N_t = N_T \cdot |M_T| = 3N_T$ multi-type genes. The blue, red, and green parts separately denote the classify, attack, and verify genes. Five elements are contained in each multi-type gene: execution order, target ID, task type, UAV ID, and UAV's heading angle. Each gene describes one configuration of a certain UAV. For example, the first gene is a classify gene representing that UAV $U_1^S$ is assigned to perform task $C$ on target $T_1$ with the heading angle of 296°.

Based on Figure 2, we introduce two transformations of the chromosome: target-based and UAV-based chromosomes.

Figure 3a can be derived from Figure 2 based on the task sequence of each target, and Figure 3b can be derived from Figure 2 based on the task executing sequence of each UAV. The original chromosome in Figure 2 can be derived from the target-based/UAV-based chromosome based on the execution order.

The introduction of the execution order is mainly used to realize the transformation from the target-based/UAV-based chromosome to the chromosome. Therefore, the chromosome, target-based chromosome, and UAV-based chromosome can be mutually transformed.

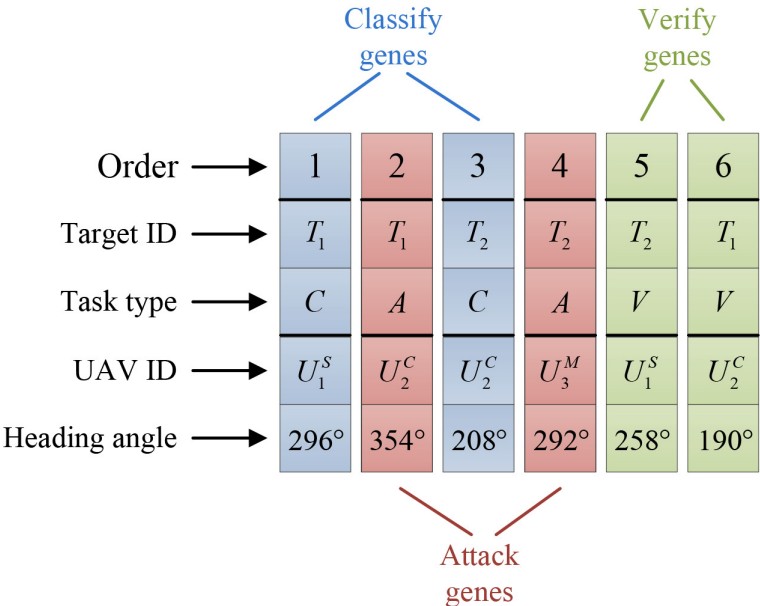

**Figure 2.** Chromosome with multi-type genes.

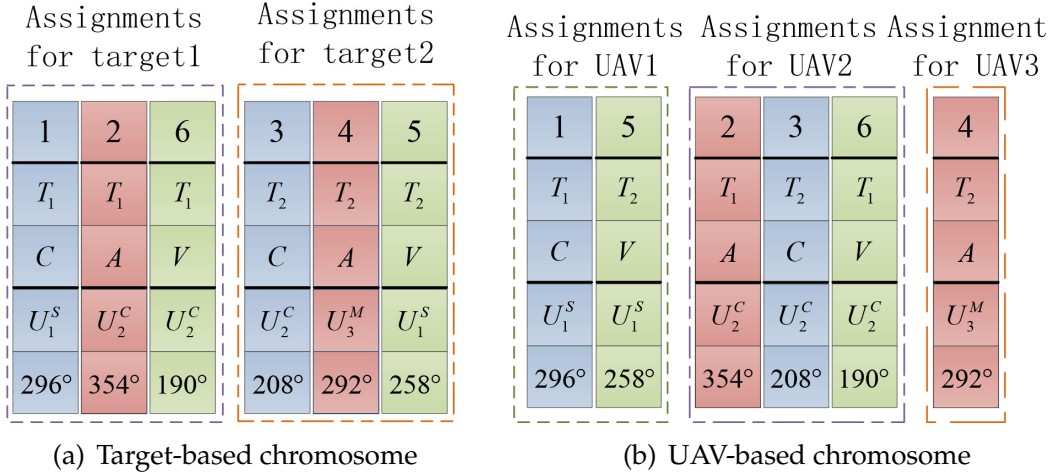

(a) Target-based chromosome      (b) UAV-based chromosome

**Figure 3.** Two transformations of the original chromosome.

We can see from Figure 3a that the feasible chromosome satisfies three constraints of the CMTAP model. Firstly, the assigned UAV (fourth row) in each gene has the corresponding capability that the task (third row) needs. Then, we can see from the execution orders of tasks $C, A, V$ that they are not only all executed, but also executed in sequence. Thus, the feasible chromosome with multi-type genes satisfies three constraints of the CMTAP model described in Section 2.3.

The UAV-based chromosome in Figure 3b shows the cruise configurations of each UAV: $U_1^S$ classifies target $T_1$ and then verifies target $T_2$ after it has been attacked by $U_3^M$. $U_2^C$ firstly attacks target $T_1$ and then flies to classify target $T_2$, and at last, verifies $T_1$. The task assignment sequence of each UAV is derived as:

$$U_1^S : (Base, \varphi_1^0) \to (T_1^C, 296°) \to (T_2^V, 258°) \tag{11}$$

$$U_2^C : (Base, \varphi_2^0) \to (T_1^A, 354°) \to (T_2^C, 208°) \to (T_1^V, 190°) \tag{12}$$

$$U_3^M : (Base, \varphi_3^0) \rightarrow (T_2^A, 292°) \tag{13}$$

The cruise configurations of UAVs are used to calculate the objective value of the CMTAP model and the fitness value of the chromosome, which will be discussed in Section 3.2.

### 3.1.2. Population Initialization

How to generate feasible chromosomes is critical for GA application. According to the proposed chromosome encoding strategy with multi-type genes, we raised the population initialization to ensure the feasibility of the GA population. Algorithm 1 shows the initialization steps to generate $N_p$ ($N_p$ is the population size) chromosomes as the initial population.

---

**Algorithm 1** Population initialization.

---

**Input T**, $N_T$, $|M_T|$, $N_t$; **U**, **U**$_S$, **U**$_A$.
**Output** $N_p$ feasible chromosomes.
Step 1: Create a $5 \times N_t$ all-zero matrix to define the chromosome.
Step 2: Generate the task execution order from one to $N_t$ as the first row of chromosome.
Step 3: Randomly arrange $N_T$ tasks $|M_T|$ times as the second row of chromosome.
Step 4: Transform the original chromosome to the target-based chromosome. For each target, add the tasks $C, A, V$ of each target successively as the third row of the target-based chromosome.
Step 5: According to the required task type in the third row, randomly select a capable UAV from the corresponding UAV set **U**$_S$/**U**$_A$ as the assigned UAV in the fourth row.
Step 6: Transform the target-based chromosome to the original chromosome. Randomly generate the heading angle as the fifth row of the chromosome.
Step 7: Repeat Steps 1-6 $N_p$ times to get the initial population.

---

In Algorithm 1, Step 1 creates $N_t = N_T \cdot |M_T| = 3N_T$ multi-type genes to construct the chromosome. Steps 2–6 separately build the five elements (execution order, target ID, task type, UAV ID, and UAV's heading angle) for each chromosome. Steps 3–4 ensure that tasks C, A, V on each target are performed in sequence. Thus, the task coupling constraints of the CMTAP model in Equations (8)–(10) are satisfied. Then, Step 5 guarantees that the assigned UAV has the corresponding capability for the required task type. Hence, the constraint on the UAVs' capability is also met.

Therefore, Algorithm 1 generates $N_p$ feasible chromosomes that follow the design of the multi-type-gene chromosome encoding scheme.

### 3.2. Calculation of Fitness

Derived from the UAV-based chromosome in Figure 3b, we can obtain the configurations of UAVs in Equations (11)–(13). According to the configurations of UAVs, the Dubins model is adopted to compute the objective value of the CMTAP model and derive the fitness value of GA.

Suppose that the cruise speeds, the turning radii, and the initial heading angles of UAVs are:

$$[v_1, v_2, v_3] = [70, 80, 70] \text{ m/s} \tag{14}$$

$$[r_1, r_2, r_3] = [200, 250, 200] \text{ m} \tag{15}$$

$$[\varphi_1^0, \varphi_2^0, \varphi_3^0] = [0°, 45°, 90°] \tag{16}$$

Based on the configurations of UAVs in Equations (11)–(13), the trajectories of UAVs are shown in Figure 4.

Then, we can derive the cruise times of UAVs based on Equation (7).

$$[t_{U_1}, t_{U_2}, t_{U_3}] = [120.3473, 162.4719, 118.0666] \text{ s} \tag{17}$$

Accordingly, the objective value of the chromosome in Figure 2 is:

$$J = \max_{U_k \in \mathbf{U}} t_{U_k} = 162.4719 \text{ s} \tag{18}$$

Hence, the feasible chromosome with multi-type genes can describe the objective function of the CMTAP model perfectly.

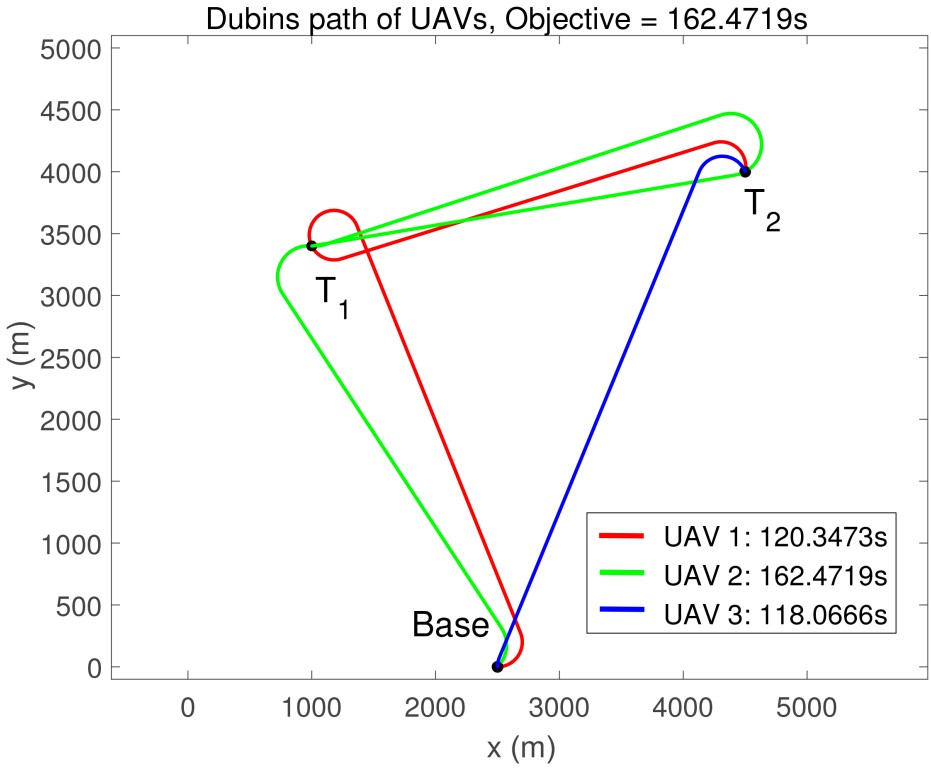

**Figure 4.** Trajectories of UAVs.

Assume that fitness values are linearly assigned between zero and one to the chromosomes of each generation based on a minimization objective, with one being the best fitness [28]. The fitness values of chromosomes are derived accordingly. For example, suppose that the GA population has $N_p = 100$ chromosomes; the fitness values of the GA population are $[0, 0.01, 0.02, \ldots, 0.99, 1]$.

*3.3. Genetic Operations*

The suitable selection, elitism, crossover, and mutation operators can not only generate feasible offspring chromosomes, but also make GA have great convergence performance. In this paper, the elitism operator is used to preserve $N_e$ parent chromosomes with the highest fitness values into the offspring population. The roulette wheel method is introduced to select parent chromosomes for the mating pool. The selection strategy with the roulette wheel method is adopted to ensure that the chromosome with a higher fitness value has a greater possibility to be chosen. The crossover and mutation operators are modified according to the chromosome encoding strategy with multi-type genes.

3.3.1. Crossover Operators

The crossover operation exchanges the gene information of two selected parent chromosomes to generate two offspring chromosomes. The purpose of the crossover operation is to improve the global searching ability through information exchange. The two-point crossover operator is used in the

proposed GA to generate $N_{cr} < N_p - N_e$ offspring chromosomes. The crossover operation is realized by Algorithm 2.

---

**Algorithm 2** Crossover operation.

---

**Input** $N_p$ parent chromosomes.
**Output** $N_{cr}$ offspring chromosomes.
Step 1: Transform two selected parent chromosomes to the target-order parent chromosomes, and randomly choose two crossover sites.
Step 2: Exchange the gene information of two target-order parent chromosomes between two crossover sites to generate two target-order offspring chromosomes.
Step 3: Transform the target-order offspring chromosomes back to offspring chromosomes.
Step 4: Repeat Steps 1-3 $N_{cr}/2$ times to generate $N_{cr}$ offspring chromosomes.

---

An example of the crossover operation is illustrated in Figure 5.

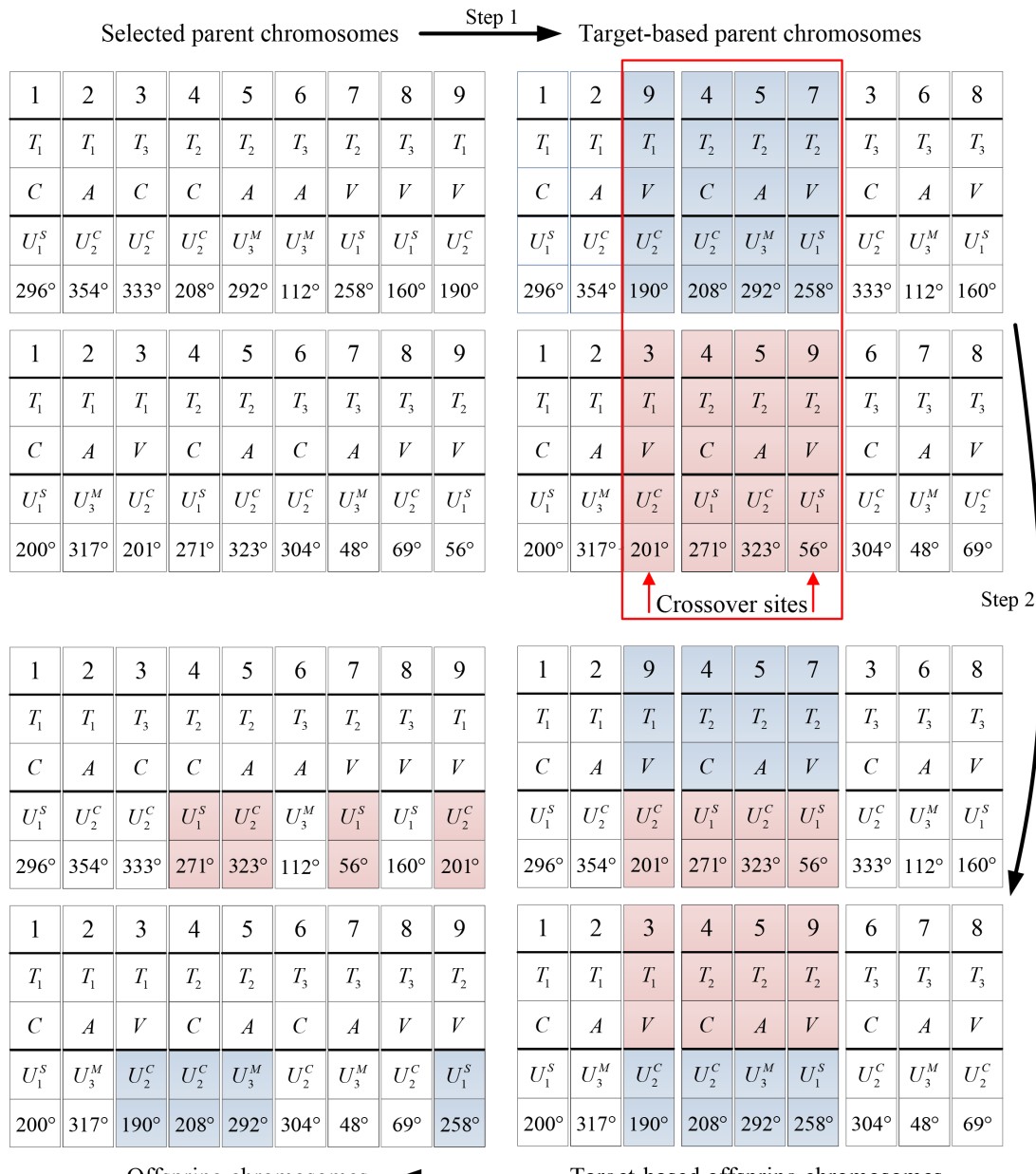

**Figure 5.** Crossover example.

We can see from Figure 5 that:

(1) The information exchange in Step 2 is performed on target-order chromosomes. As the second to third rows of two parent target-order chromosomes are the same, the crossover operation only exchanges the assigned UAVs and corresponding heading angles of selected genes. Thus, the offspring chromosomes generated by the crossover operation still satisfy the constraint in Equations (8) and (9) that tasks *C*, *A*, *V* are all performed.

(2) The first row is not involved in the crossover operation because the first row realizes the transformation from the target-based chromosome to the chromosome. At the same time, the unchanged execution order ensures that the offspring chromosomes generated by the crossover operation still satisfy the task precedence constraint in Equation (10).

(3) For two target-based parent chromosomes, the target and task information of the selected crossover sites are the same. Hence, the offspring chromosomes generated by the crossover operation will not violate the constraint on the UAVs' capability.

According to the above analyses, as long as the parent chromosomes satisfy the constraints of the CMTAP model, the crossover operation can generate feasible offspring chromosomes. Therefore, the proposed crossover operation is effective for the CMTAP model.

### 3.3.2. Mutation Operators

The mutation operation changes one or more genes of the selected parent chromosome to generate the offspring chromosome. The objective of the mutation operation is to increase the local searching ability through the disturbance of the gene(s). Four mutation operators are put forward to generate $N_{mu} = N_p - N_e - N_{cr}$ offspring chromosomes. two types of mutation operators are put forward in this paper.

(a) Mutation of the assigned information: Two mutation operators are utilized. Randomly select the mutation site, and mutate the assigned UAV or assigned heading angle of the mutation site. We can get two mutated offspring chromosomes. The implementation examples are shown in Figure 6.

Selected parent chromosome

| 1 | 2 | 3 | 4 | 5 | 6 | 7 | 8 | 9 |
|---|---|---|---|---|---|---|---|---|
| $T_1$ | $T_1$ | $T_3$ | $T_2$ | $T_2$ | $T_3$ | $T_2$ | $T_3$ | $T_1$ |
| $C$ | $A$ | $C$ | $C$ | $A$ | $A$ | $V$ | $V$ | $V$ |
| $U_1^S$ | $U_2^C$ | $U_2^C$ | $U_2^C$ | $U_3^M$ | $U_3^M$ | $U_1^S$ | $U_1^S$ | $U_2^C$ |
| 296° | 354° | 333° | 208° | 292° | 112° | 258° | 160° | 190° |

Mutation site

| 1 | 2 | 3 | 4 | 5 | 6 | 7 | 8 | 9 |
|---|---|---|---|---|---|---|---|---|
| $T_1$ | $T_1$ | $T_3$ | $T_2$ | $T_2$ | $T_3$ | $T_2$ | $T_3$ | $T_1$ |
| $C$ | $A$ | $C$ | $C$ | $A$ | $A$ | $V$ | $V$ | $V$ |
| $U_1^S$ | $U_2^C$ | $U_2^C$ | $U_1^S$ | $U_3^M$ | $U_3^M$ | $U_1^S$ | $U_1^S$ | $U_2^C$ |
| 296° | 354° | 333° | 208° | 292° | 112° | 258° | 160° | 190° |

(a) Mutate the assigned UAV

| 1 | 2 | 3 | 4 | 5 | 6 | 7 | 8 | 9 |
|---|---|---|---|---|---|---|---|---|
| $T_1$ | $T_1$ | $T_3$ | $T_2$ | $T_2$ | $T_3$ | $T_2$ | $T_3$ | $T_1$ |
| $C$ | $A$ | $C$ | $C$ | $A$ | $A$ | $V$ | $V$ | $V$ |
| $U_1^S$ | $U_2^C$ | $U_2^C$ | $U_2^C$ | $U_3^M$ | $U_3^M$ | $U_1^S$ | $U_1^S$ | $U_2^C$ |
| 296° | 354° | 333° | 56° | 292° | 112° | 258° | 160° | 190° |

(b) Mutate the assigned heading angle

**Figure 6.** Mutation of assigned information.

We can see from Figure 6a that the assigned UAV of the mutation site is mutated. The UAV sets with different capabilities are $\mathbf{U}_S = \{U_1^S, U_2^C\}, \mathbf{U}_A = \{U_2^C, U_3^M\}$. Thus, the mutated UAV of the selected site should be $\mathbf{U}_S \setminus \{U_2^C\} = \{U_1^S\}$. If there are multiple capable UAVs in the corresponding UAV set, we randomly select the mutated UAV.

Figure 6b shows that the assigned heading angle of selected gene is mutated. As the fitness value of the chromosome is derived from the Dubins paths of UAVs, the mutation of the assigned heading angle can explore more possible chromosomes in the solution space.

(b) Mutation based on state-transition strategy: Each chromosome is regarded as a state of the solution space. The GA iterations are recognized as the state transition process. Accordingly, a state-transition vector is adopted in the mutation processing to explore the solution space. The mutation operators with the state-transition vector have a better stochastic characteristic, which can help the algorithm jump out of the local optimal solution and enhance the diversity of the GA population.

An example of the state transition of the three elements is exhibited in Figure 7.

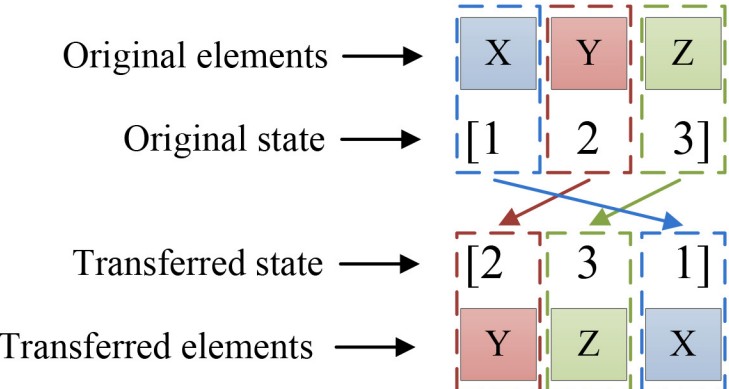

**Figure 7.** State transition of three elements.

In Figure 7, the original state $[1, 2, 3]$ representing elements $[X, Y, Z]$ is transferred to $[2, 3, 1]$ representing elements $[Y, Z, X]$. There are $(n! - 1)$ potential states for the transferred elements. For example, the original state is $[1, 2, 3]$. The possible transferred states of three elements are:

$$[1, 3, 2], [2, 1, 3], [2, 3, 1], [3, 1, 2], [3, 2, 1] \tag{19}$$

Randomly select one parent chromosome. The state-transition strategy is firstly used to exchange the execution order of the targets. Taking the task assignment sequence of each target as a unit, the state-transition vector is used to exchange the execution order of the targets. The number of targets is $N_T$; thus, the number of potential transferred states is $(N_T! - 1)$.

Firstly, we randomly generate the state-transition vector for the selected parent target-order chromosome. Then, the corresponding genes are exchanged. An implementation example is given in Figure 8. The random state-transition vector is $[2, 3, 1]$.

The mutation processing in Figure 8 only exchanges the execution order of the targets. Thus, the offspring chromosome still meet the constraints of the CMTAP model.

Besides, another mutation operator based on the state-transition strategy is raised to exchange the assigned information of the selected task. Randomly select the mutated task and the state-transition vector. The state-transition vector is used to exchange the assigned information of the selected task. The assigned information of the selected task is regarded as the unit; thus, the number of potential transferred states is $(|M_T|! - 1)$. An implementation example is exhibited in Figure 9. The random state-transition vector is $[2, 3, 1]$.

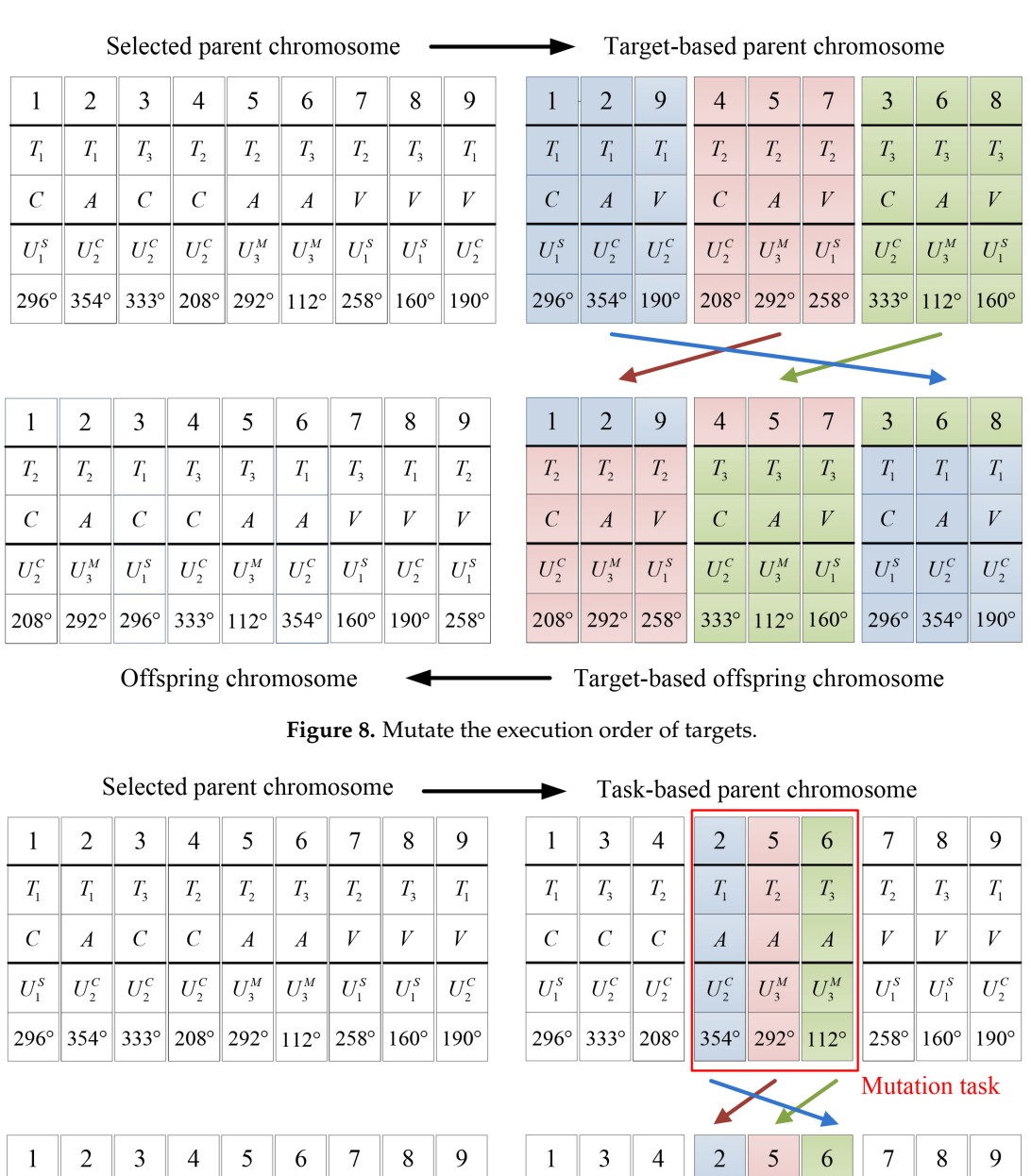

**Figure 8.** Mutate the execution order of targets.

**Figure 9.** Mutate the assigned information of a certain task.

The mutation processing in Figure 9 only exchanges the assigned information of the selected task. Thus, the offspring chromosome will not violate the constraints of the CMTAP model.

The proposed AGA's four mutation operators are shown in Figures 6, 8, and 9, and Algorithm 3 shows the implementation of the mutation operation that generates $N_{mu}$ offspring chromosomes.

---

**Algorithm 3** Mutation operation.

---

**Input** $N_p$ parent chromosomes.
**Output** $N_{mu}$ offspring chromosomes.
Step 1: Randomly select the parent chromosome, and transform the parent chromosome to the target-order parent chromosome.
Step 2: Randomly apply one mutation way to generate the target-order offspring chromosome.
Step 3: Transform the target-order offspring chromosomes back to the offspring chromosome.
Step 4: Repeat Steps 1-3 $N_{mu}$ times to complete the mutation operation.

---

### 3.4. Adaptive Setting

Finally, the idea of the adaptive setting in the genetic operations is raised. The proposed AGA will dynamically adjust the number of crossover and mutation offspring chromosomes along with the iteration time.

The offspring population consists of $N_e$ chromosomes generated by the elitism operation, $N_{cr}$ chromosomes generated by the crossover operation, and $N_{mu}$ chromosomes generated by the mutation operation. In order to solve the CMTAP model with the proposed AGA, we firstly need to determine these parameters. The subjective selection of parameters $N_e, N_{cr}, N_{mu}$ will affect the performance of GA. To avoid this issue, the proposed GA dynamically adjusts parameters $N_{cr}, N_{mu}$ according to the iteration time.

$$N_{cr} = round[(N_p - N_e) \cdot e^{(-\frac{N_{iter}}{N_g})}] \tag{20}$$

$$N_{mu} = N_p - N_e - N_{cr} \tag{21}$$

where $N_{iter}, N_g$ separately represent the current iteration time and total iteration time and $round[\cdot]$ is the rounding function.

Let $N_p = 100, N_e = 4$, and $N_g = 300$. The dynamic adjustments of $N_{cr}, N_{mu}$ are shown in Figure 10.

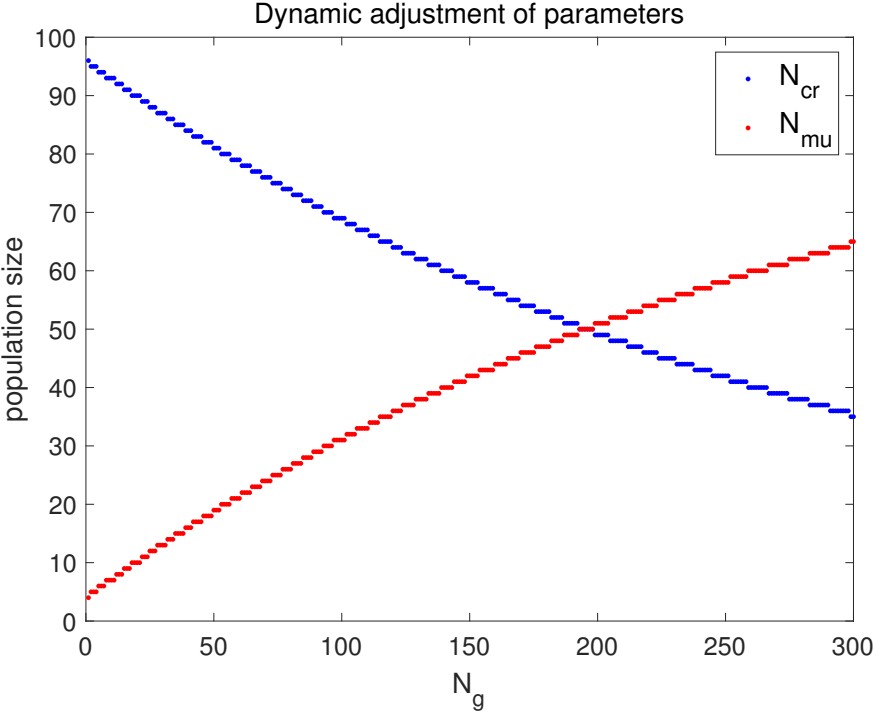

**Figure 10.** Adaptive setting of $N_{cr}, N_{mu}$.

We can see from Figure 10 that:

(1) At the beginning stage of the iterations, the crossover population is relatively large. Thus, the information exchanges of the crossover operation are conducive to enhancing the global searching ability of AGA.

(2) At the later stage of iterations, the mutation population is relatively large. Hence, the refined searching of the mutation operation can accelerate the convergence speed of AGA.

Therefore, dynamic parameter settings can better balance the global and local searching ability during the genetic iterations.

## 4. Simulations and Analyses

An adaptive genetic algorithm is proposed in this paper to solve the cooperative multiple task assignment for heterogeneous UAVs. To prove the effectiveness and superiority of the proposed AGA, one-hundred Monte Carlo simulations were performed. In the interest area of a 5 km × 5 km square, a multi-UAV team from the same base needs to perform the SEAD mission on multiple stationary ground targets. The duration of each task ($C/A/V$) is 5 s.

To demonstrate the performance of the proposed AGA comprehensively, the random search method (RS), GA, ACO, and PSO were chosen as the comparison algorithms. RS represents the optimization method with the random searching strategy. ACO and PSO represent the swarm intelligent optimization methods. Hence, RS, ACO, and PSO were introduced to verify that GA and AGA had better searching ability when solving the CMTAP model. Besides, GA was used to prove that the proposed AGA with the adaptive setting (in Section 3.4) could obtain better optimization results. The simulation parameters of these methods are given in Table 1.

**Table 1.** Simulation parameters of the methods. RS, random search; AGA, adaptive genetic algorithm.

| Methods | Simulation Parameters |
|---|---|
| RS | $N_p = 100$ |
| GA | $N_p = 100, N_e = 4, N_{cr} = 66, N_{mu} = 30$ |
| AGA | $N_p = 100, N_e = 4$ |
| ACO | $N_{ant} = 100, \alpha = 1, \beta = 0.5, \rho = 0.1, Q = 1$ |
| PSO | $N_{particle} = 1000, c_1 = 0.5, c_2 = 0.7$ |

The iteration time of all methods was set as $N_g = 300$. In Table 1, $N_{ant}, \alpha, \beta, \rho, Q$ of ACO respectively denote the number of ants, the influence factor of pheromone density, the influence factor of heuristic information, the pheromone evaporation coefficient, and the pheromone intensity. $N_{particle}, c_1, c_2$ separately define the number of particles, the acceleration coefficient of personal cognition, and the acceleration coefficient of social cognition.

### 4.1. Feasibility of the Proposed Algorithm

In Section 3.1.1, we give the example scenario that the multi-UAV system $\mathbf{U} = \{U_1^S, U_2^C, U_3^M\}$ from the same base needs to perform the SEAD mission on targets $\mathbf{T} = \{T_1, T_2\}$. According to Equations (14)–(17), the simulation parameters are shown in Table 2.

**Table 2.** Parameter settings.

| | |
|---|---|
| UAV-related | Cruise speeds $[v_1, v_2, v_3] = [70, 80, 70]$ m/s<br>Turning radii $[r_1, r_2, r_3] = [200, 250, 200]$ m<br>Initial heading angles $[\varphi_1, \varphi_2, \varphi_3] = [0°, 45°, 90°]$ |
| Coordinates | Base $[2500, 0]$ m<br>Targets $[1000, 3400; 4500, 4000]$ m |

Using the proposed AGA, we can get the task assignment results of UAVs.

$$U_1^S : (Base, \varphi_1^0) \rightarrow (T_2^C, 340°) \rightarrow (T_2^V, 17°) \tag{22}$$

$$U_2^C : (Base, \varphi_2^0) \rightarrow (T_1^C, 128°) \rightarrow (T_1^A, 131°) \rightarrow (T_1^V, 133°) \tag{23}$$

$$U_3^M : (Base, \varphi_3^0) \rightarrow (T_2^A, 245°) \tag{24}$$

We can see from the task schedules in Equations (22)–(24) that the task assignment results satisfied the constraint on UAVs' different capabilities, e.g., $U_1^S$ was a surveillance UAV that could perform the classify and verify tasks in the SEAD mission. In Equation (22), only tasks $C, V$ are assigned to $U_1^S$. Accordingly, the arranged trajectories of UAVs obtained by the proposed algorithm are shown in Figure 11.

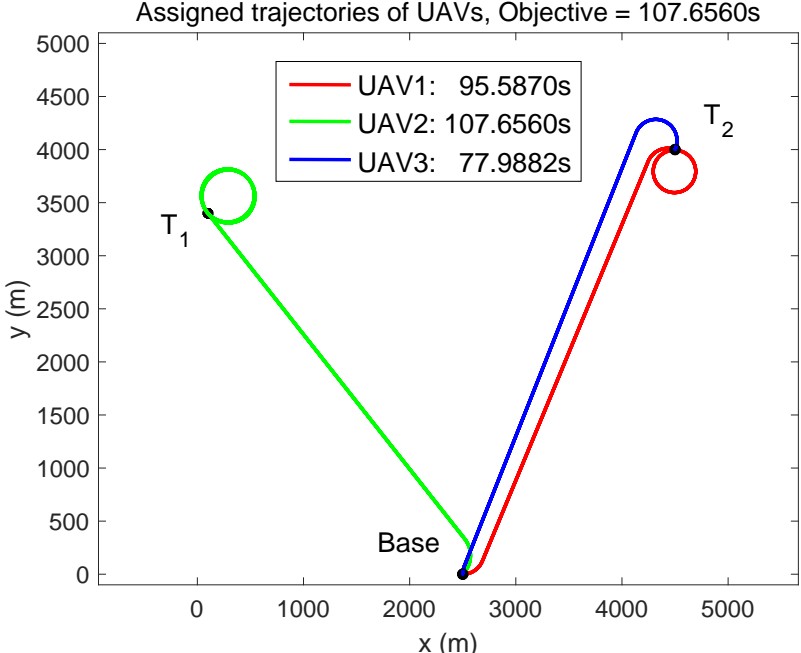

**Figure 11.** Trajectories of UAVs.

Figure 4 shows one potential task schedule of UAVs. Compared with Figure 4, the arranged trajectories of UAVs obtained by the proposed algorithm in Figure 11 had a lower time cost. That is, the proposed algorithm produced a feasible task assignment result that guaranteed that the multi-UAV system could complete the SEAD mission within a short time.

*4.2. Scenario 1: Three UAVs against Four Targets*

We discussed the scenario of the multi-UAV team $\mathbf{U} = \{U_1^S, U_2^C, U_3^M\}$ performing the SEAD mission on four targets. Referring to [23], the parameters of Scenario 1 are shown in Table 3.

**Table 3.** Parameters of Scenario 1.

| | |
|---|---|
| UAV-related | Cruise speeds $[v_1, v_2, v_3] = [70, 80, 70]$ m/s<br>Turning radii $[r_1, r_2, r_3] = [200, 250, 200]$ m<br>Initial heading angles $[\varphi_1, \varphi_2, \varphi_3] = [0°, 45°, 90°]$ |
| Coordinates | Base $[2500, 0]$ m<br>Targets $[800, 3200; 2800, 4200; 4000, 2000; 1000, 760]$ m |

The results of 100 Monte Carlo simulations are shown in Figure 12.

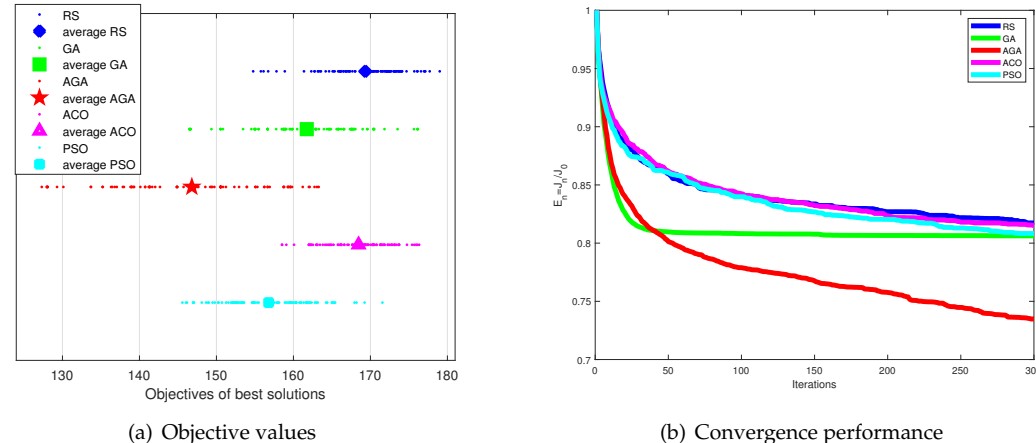

(a) Objective values　　　　　　　　　　　　　　(b) Convergence performance

**Figure 12.** Monte Carlo results of Scenario 1.

Figure 12a reveals the objective results of the five methods within 100 Monte Carlo simulations. The smaller the objective value was, the better optimization effect the algorithm had. It can be seen from the distribution of the objective values that the proposed AGA had lower objective values than other methods. Thus, the proposed AGA had a better optimization ability compared with RS, GA, ACO, and PSO.

Figure 12b gives the convergence performance of the five methods within 100 Monte Carlo simulations. The convergence index $E_n = \frac{J_n}{J_0}$ was adopted to describe the convergence effect. $J_n$ is the objective value after $n$ iterations, and $J_0$ is the initial objective value before iterations. Hence, the smaller $E_{N_g}$ is, the better the convergence performance the algorithm has. We can see from Figure 12b that the proposed AGA had lower $E_n$ than other methods after 50 iterations. Obviously, the proposed AGA had better convergence performance.

### 4.3. Scenario 2: Five UAVs against Nine Targets

Then, we conducted the simulation of the multi-UAV team $\mathbf{U} = \{U_1^S, U_2^C, U_3^C, U_4^C, U_5^M\}$ performing the SEAD mission on nine targets. The parameters are shown in Table 4.

**Table 4.** Parameters of Scenario 2.

| | |
|---|---|
| | Cruise speeds $[v_1, v_2, v_3] = [70, 80, 70, 90, 60]$ m/s |
| UAV-related | Turning radii $[r_1, r_2, r_3] = [200, 250, 200, 300, 180]$ m |
| | Initial heading angles $\begin{array}{c}[\varphi_1, \varphi_2, \varphi_3, \varphi_4, \varphi_5]\\ = [0°, 22.5°, 45°, 67.5°, 90°]\end{array}$ |
| | Base $[0, 0]$ m |
| Coordinates | Targets $\begin{bmatrix} 300, 400; 4350, 900; 1500, 450; \\ 3850, 650; 3900, 4700; 2750, 150; \\ 1000, 2750; 4750, 4300; 5000, 5000 \end{bmatrix}$ m |

Accordingly, the Monte Carlo results of Scenario 2 are exhibited in Figure 13.

We can see from Figure 13 that PSO had bad performance in Scenario 2. The work in [19] revealed that PSO may produce a larger number of infeasible particles during the population iterations. The infeasible particles affect the optimization searching ability of PSO badly.

In Figure 13, GA and the proposed AGA had better performance than RS, ACO, and PSO. Apparently, the unique designs of the chromosome encoding strategy and genetic operators were conducive to producing better CMTAP solutions. Further, the AGA had better performance than GA. Hence, the adaptive setting of AGA allowed a dynamic searching of the feasible solution space, which was more reliable to obtain better CMTAP solutions.

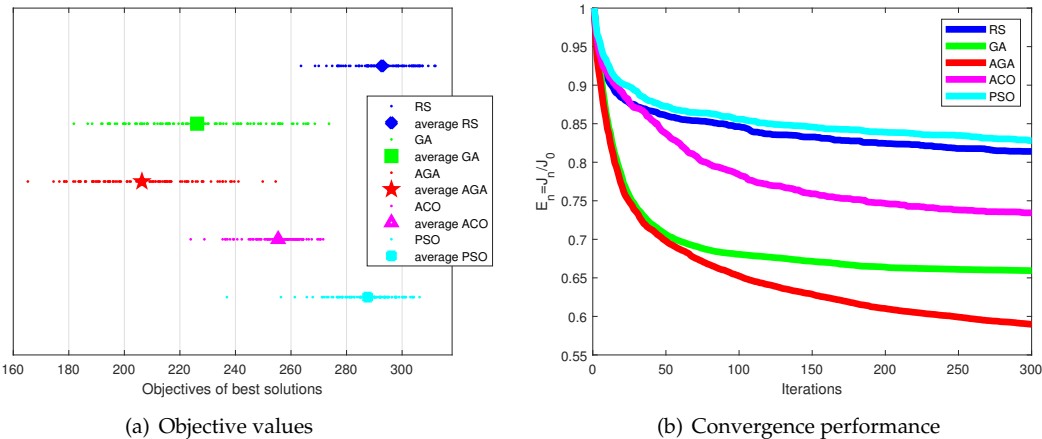

(a) Objective values  (b) Convergence performance

**Figure 13.** Monte Carlo results of Scenario 2.

Accordingly, the proposed AGA had lower objective values and a faster convergence effect than RS, GA, ACO, and PSO. That is, the proposed AGA had better optimization ability and convergence performance than RS, GA, ACO, and PSO.

### 4.4. Scenario 3: 15 UAVs against 10 Targets

Increasing the scale size of the CMTAP model, we discussed the scenario of the multi-UAV team $\mathbf{U} = \{U_1^S, \ldots, U_5^S, U_6^C, \ldots, U_{10}^C, U_{11}^M, \ldots, U_{15}^M\}$ performing the SEAD mission on 10 targets. Referring to [27], the parameters are shown in Table 5.

**Table 5.** Parameters of Scenario 3.

| UAV-related | Range of random cruise speeds: $[50-100]$ m/s |
| | Range of random cruise speeds: $[50-100]$ m/s |
| | Range of random initial heading angles: $[150-300]$ m |
| Coordinates | Base $[2500,0]$ m |
| | Random 10 targets in the interest area |

The Monte Carlo results of Scenario 3 are exhibited in Figure 14.

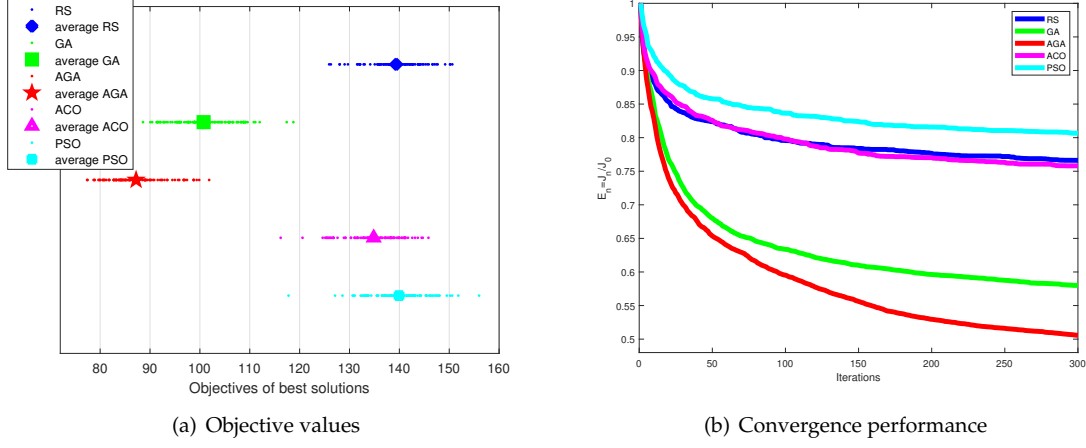

(a) Objective values  (b) Convergence performance

**Figure 14.** Monte Carlo results of Scenario 3.

In Figure 14, the proposed AGA had lower objective values compared with other methods. Then, the proposed AGA had the lowest convergence index. Thus, Figure 14 further proves the effectiveness and superiority of the proposed AGA.

Besides, compared with the simulation results of Scenarios 1–2 in Figures 13 and 14, the GA and proposed AGA in Scenario 3 had obviously better optimization performance that other algorithms. Hence, the raised crossover and mutation operators made GA and AGA have better optimization ability, especially in a large scale scenario. Simultaneously, the proposed AGA had better objective values and convergence performance than GA. Thus, the adaptive setting of AGA made the proposed algorithm have stronger dynamic searching ability to get better optimization performance than GA.

*4.5. Discussions*

To elaborate the optimization ability and convergence effect of the proposed AGA, the results of the above three scenarios are given in Table 6. The optimization indexes of the five methods are reflected by two aspects. Firstly, the min, max, and average numbers of the objective values within 100 Monte Carlo simulations are shown. Then, the convergence index $E_n = \frac{J_n}{J_0}$ is adopted.

**Table 6.** Monte Carlo results of the five methods in 3 scenarios.

| Scenarios | Optimization Results (min, max, avg, $E_n$) | | | | |
|---|---|---|---|---|---|
| | **AGA** | **GA** | **RS** | **ACO** | **PSO** |
| 3 UAVs vs 4 Targets | **127.31** | 146.50 | 154.77 | 158.52 | 145.58 |
| | **163.28** | 176.16 | 179.00 | 176.33 | 171.56 |
| | **146.81** | 161.73 | 169.34 | 168.47 | 156.78 |
| | **0.7348** | 0.8063 | 0.8174 | 0.8150 | 0.8083 |
| 5 UAVs vs 9 targets | **165.25** | 181.81 | 263.60 | 223.88 | 236.92 |
| | **254.48** | 273.68 | 311.98 | 271.58 | 306.29 |
| | **206.33** | 226.21 | 292.80 | 255.38 | 287.58 |
| | **0.5897** | 0.6594 | 0.8140 | 0.7344 | 0.8280 |
| 15 UAVs vs 10 targets | **77.38** | 88.59 | 125.92 | 116.20 | 117.77 |
| | **101.86** | 118.75 | 150.62 | 145.83 | 155.98 |
| | **87.21** | 100.75 | 139.36 | 134.85 | 139.93 |
| | **0.5057** | 0.5799 | 0.7661 | 0.7580 | 0.8065 |

The best results of each optimization index are highlighted in bold. We can see from Table 6 that:

(1) From the distribution of objective values within 100 Monte Carlo simulations, the proposed AGA had the lowest min/max/avg values. Thus, the proposed AGA could obtain better CMTAP solutions compared with RS, GA, ACO, and PSO under different scenarios.

(2) The convergence indexes $E_n$ of RS, ACO, and PSO under different scenarios were always within $[0.73 - 0.82]$. With the increasing of the scenario scale, the convergence indexes of GA and the proposed AGA decreased gradually. Obviously, the designs of chromosome encoding strategy and corresponding genetic operators helped GA and the proposed AGA obtain a better convergence effect under complex scenarios.

(3) The convergence indexes of the proposed AGA were always smaller than that of GA, e.g., the convergence index of GA decreased at around 0.58 in Scenario 3, while the convergence index of the proposed AGA decreased at around 0.51. Obviously, the design of the adaptive setting made the proposed AGA obtain a better convergence effect than GA.

The simulation results of the above three scenarios discussed the performance of the proposed AGA compared with RS, GA, ACO, and PSO. The analyses above verified that the proposed AGA had better optimization ability and convergence performance under different CMTAP scales. Therefore, the effectiveness and superiority of the proposed AGA were proven.

## 5. Conclusions

In this paper, we discussed the cooperative multiple task assignment problem (CMTAP) in an SEAD mission. The multi-UAV team needed to perform the classify, attack, and verify tasks consecutively on multiple stationary ground targets. To solve CMTAP, we proposed an adaptive genetic algorithm (AGA) with a multi-type gene chromosome encoding strategy. On the one hand, the proposed AGA designed a unique chromosome encoding strategy to generate feasible chromosomes that satisfied the UAVs' heterogeneity and task coupling constraints. On the other hand, the applicable crossover and mutation operators were designed to enhance the optimization ability and convergence effect of the algorithm. Besides, the proposed AGA dynamically adjusted the population numbers of the crossover and mutation operations. The simulation results proved the feasibility and effectiveness of the proposed AGA for CMTAP under different scenarios.

In actual applications, there may be more constraints on the CMTAP model, e.g., UAVs may have limited capacities, and tasks may need to be performed in certain time windows. Thus, we will concentrate on how to apply the proposed AGA to solve the CMTAP with complex constraints in future research. Besides, the proposed AGA produced the preset trajectories for UAVs to perform its assigned tasks. In practical applications, UAVs need to adjust their cruise path according to environmental conditions (e.g., obstacles, complex terrain, dynamic environment). Therefore, we also need to study the online path adjustment strategies for UAVs.

**Author Contributions:** J.C. and F.Y. conceived of the concept and performed the research. J.C. conducted the simulations and wrote the manuscript. Y.T. and T.J. reviewed the manuscript. All authors read and approved the final manuscript.

**Funding:** The paper is funded by the National Natural Science Foundation of China (No. 61701134, No. 51809056), the National Key Research and Development Program of China (No. 2016YFF0102806), and the Natural Science Foundation of Heilongjiang Province, China (No. F2017004).

**Conflicts of Interest:** The authors declare no conflict of interest.

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
