# Peer review of "Cooperative Task Assignment of a Heterogeneous Multi-UAV System Using an Adaptive Genetic Algorithm"

_electronics, doi:10.3390/electronics9040687_

Round 1

Reviewer 1 Report

This paper describes a method that using genetic algorithms deals with cooperative task assignments, applied to a heterogeneous set of UAVs. The proposed scenario is a SEAD mission with multiple ground targets.

The paper is very well written and structured, with all the concepts clearly exposed and explained, and the application it describes is fairly interesting, not only in the proposed scenario but regarding its extension to others.

Nevertheless, there are some issues that must be fixed:

  • Eq. 1 and Eq. 3 both include a parameter identified as Number of Targets, but both names are different. Clarify the difference between them or use the same name. From Eq 4 and Eq 5 it seems that N_u represents the Number uf UAVs, not Number of Targets
  • The objective function is somehow too simple; minimum time forces UAVs to fly full throttle, shortening the autonomy. Why this choice, then? Moreover, in Eq. 7 v_k seems to represent a constant speed for the whole mission, regardless the mission circumstances; why this constancy (if v_k is not v_kmax)?
  • Besides conditions depicted in Eqs 8 and 9, there must be another one forcing that \Sum_{k=1}^{N_u} X^{U_k,m}_{(q_i,q_j)}=1, forcing every mission to be assigned and preventing that any mission is assigned more than once
  • Could it be possible to show the trajectories of UAVs in a simple case in the Simulation and Analyses section?

Author Response

Dear Reviewers:

I want to express my very deep appreciation, and the appreciation of all of us, to your great efforts and suggestions for our manuscript. They are valuable and very helpful for revising and improving our paper, as well as the important guiding to our researches.

  • The English usage has been careful edited by authors and one native English speaker.
  • The whole article has been revised according to the reviewers’ comments.
  • The primary modifications are highlighted in the revised version.
  • The point-to-point responses to the comments are shown as follows.

------------------------------------------------------------------------------------------

  1. Eq. 1 and Eq. 3 both include a parameter identified as Number of Targets, but both names are different. Clarify the difference between them or use the same name. From Eq 4 and Eq 5 it seems that N_u represents the Number uf UAVs, not Number of Targets

Reply:Sorry about the mistake. N_T in Eq.1 represents the number of targets, and N_U in Eq.3 represents the number of UAVs. We have gone through the whole manuscript, and corrected the corresponding errors, please check.

------------------------------------------------------------------------------------------

  1. The objective function is somehow too simple; minimum time forces UAVs to fly full throttle, shortening the autonomy. Why this choice, then? Moreover, in Eq. 7 v_k seems to represent a constant speed for the whole mission, regardless the mission circumstances; why this constancy (if v_k is not v_kmax)?

Reply:Please understand that Eqs.6-7 denote the objective function of CMTAP model. In the CMTAP problem, there are two kinds of team objectives: minimizing total cost or minimizing maximum cost. Considering that the multi-UAV system needs to perform all tasks as quickly as possible in the SEAD mission, we choose “minimizing maximum time cost” as the objective function in this paper. Thus, the task assignment result obtained by the proposed algorithm ensures that the multi-UAV system performs SEAD mission as quickly as possible.

In this paper, the cruise speeds of UAVs are consumed to be constant. Please understand that the actual flight of UAV is related to many factors such as altitude, speed, distance, wind speed, etc., thereby the actual flight path of UAV is difficult to be calculated accurately in the simulation. the proposed algorithm in this paper is used to produce the preset task schedules for UAVs. Thus, we set constant speeds for UAVs. Besides, please check Refs.[19-20,22]. The cruise speeds of UAVs are all set as constant, and the objective is to minimize the maximum cost of multi-UAV system.

------------------------------------------------------------------------------------------

  1. Besides conditions depicted in Eqs 8 and 9, there must be another one forcing that \Sum_{k=1}^{N_u} X^{U_k,m}_{(q_i,q_j)}=1, forcing every mission to be assigned and preventing that any mission is assigned more than once

Reply:Thank you for your suggestion. We add Eq.8 and corresponding description in the revised version to express that each task is assigned just once. Your kind advice makes the description on CMTAP model more precise.

------------------------------------------------------------------------------------------

  1. Could it be possible to show the trajectories of UAVs in a simple case in the Simulation and Analyses section?

Reply:In case readers don’t understand the trajectories of UAVs obtained by the proposed algorithm, we have shown the trajectories of UAVs in Figure 4. Besides, we add Figure 11 in Sect.3.1 to show the trajectories of UAVs after using the proposed algorithm, please check Pages 13-14. Compared with the potential task schedules in Figure 4, the task schedule obtained by the proposed algorithm has lower time cost. Thus, we simply proved that the proposed algorithm can produce feasible task assignment result. Thanks for your kind suggestion.

------------------------------------------------------------------------------------------

Reviewer 2 Report

It was a pleasure to review the work "Cooperative Task Assignment of Heterogeneous Multi-UAV System Using an Adaptive Genetic Algorithm". The authors presented an extensive theoretical introduction, conducted simulations of the presented methods. 

The only thing that could be improved is:
- the motivation to use the chosen methods can be extended,
- you can add a few words about competitive systems and what new features the presented one has,
- The interpretation of the results obtained should be extended,
- Given the fact that the authors present a theory and simulation, one can mention the problem of the reality gap. That is, how far the simulation may differ from the actual performance of tasks.

The overall impression is good, the work has been prepared with care, after introducing the above descriptions will be acceptable for publishing.

Author Response

Dear Reviewers:

I want to express my very deep appreciation, and the appreciation of all of us, to your great efforts and suggestions for our manuscript. They are valuable and very helpful for revising and improving our paper, as well as the important guiding to our researches.

  • The English usage has been careful edited by authors and one native English speaker.
  • The whole article has been revised according to the reviewers’ comments.
  • The primary modifications are highlighted in the revised version.
  • The point-to-point responses to the comments are shown as follows.

-------------------------------------------------------------------------

  1. - the motivation to use the chosen methods can be extended,

- you can add a few words about competitive systems and what new features the presented one has,

Reply:The motivation of this paper is to study the adaptive genetic algorithm to solve the cooperative multiple task assignment problem (CMTAP) with task coupling constraints in heterogeneous multi-UAV system. In the “Introduction”, we described the CMTAP as a NP-hard combinatorial optimization problem. Thus, intelligent optimization methods are widely used to solve the CMTAP model. Genetic algorithm has strong universality, simple encoding strategy and genetic operators, which is perfect for CMTAP model to generate feasible task assignment. Besides, considering the heterogeneity of UAVs and task coupling constraints as the features in the SEAD mission, the existent methods cannot produce feasible task assignment results. Thus, the adaptive genetic algorithm is proposed in this paper. Please check Page 1-2.

-------------------------------------------------------------------------

  1. - The interpretation of the results obtained should be extended,

Reply:The “Simulations and Analyses” has been revised. Firstly, we added the description on why we choose RS, GA, ACO and PSO as comparing methods at Page 13. Then, Sect.3.1-Feasibility of the proposed algorithm is added to show the arranged trajectories of UAVs obtained by the proposed algorithm. Please check Pages 13-14. Finally, the revised manuscript adds necessary interpretation of the simulation results, please check.

-------------------------------------------------------------------------

  1. - Given the fact that the authors present a theory and simulation, one can mention the problem of the reality gap. That is, how far the simulation may differ from the actual performance of tasks.

Reply:This paper studies the cooperative multiple task assignment problem of heterogeneous UAVs with task coupling constraints. In actual applications, more complex constraints should be considered. Besides, the proposed AGA produces preset trajectories for UAVs. UAVs need to dynamically adjust their online path according to environmental changes. Corresponding descriptions are added in “Conclusions” at Page 18, please check.

-------------------------------------------------------------------------